# Continuous Heatmap Regression for Pose Estimation via Implicit Neural Representation

**Shengxiang Hu[1], Huaijiang Sun[1]\*, Dong Wei[1], Xiaoning Sun[1], Jin Wang[2]**
[1]Nanjing University of Science and Technology, Nanjing, China
[2]Nantong University, Nantong, China
{hushengxiang,sunhuaijiang}@njust.edu.cn

## Abstract

Heatmap regression has dominated human pose estimation due to its superior performance and strong generalization. To meet the requirements of traditional explicit neural networks for output form, existing heatmap-based methods discretize the originally continuous heatmap representation into 2D pixel arrays, which leads to performance degradation due to the introduction of quantization errors. This problem is significantly exacerbated as the size of the input image decreases, which makes heatmap-based methods not much better than coordinate regression on low-resolution images. In this paper, we propose a novel neural representation for human pose estimation called NerPE to achieve continuous heatmap regression. Given any position within the image range, NerPE regresses the corresponding confidence scores for body joints according to the surrounding image features, which guarantees continuity in space and confidence during training. Thanks to the decoupling from spatial resolution, NerPE can output the predicted heatmaps at arbitrary resolution during inference without retraining, which easily achieves sub-pixel localization precision. To reduce the computational cost, we design progressive coordinate decoding to cooperate with continuous heatmap regression, in which localization no longer requires the complete generation of high-resolution heatmaps. The code is available at https://github.com/hushengxiang/NerPE.

## 1 Introduction

Human pose estimation (HPE) is a fundamental task in the field of computer vision, which is widely used in various human-centered applications [41, 29, 37, 9, 55, 53]. In multi-person pose estimation, the top-down framework is a mainstream two-stage pipeline, in which the person areas are firstly cropped out and then the body joints within them are located. According to the way to describe the positions of body joints, pose estimation methods can be further divided into coordinate regression [44, 43, 20, 22, 28] and heatmap regression [31, 48, 42, 24, 52]. Due to better performance brought by spatial encoding of heatmap representation, heatmap regression has received more attention than coordinate regression. Recently, most heatmap-based methods [11, 51, 50] focus on the design of network structure but ignore the importance of heatmap generation. Only a few works [26, 23] have noticed the irrationality in the standard heatmap representation widely used in existing methods.

In the top-down framework, the ground-truth coordinates of body joints in the original image are mapped to the input plane of a keypoint detector, through the same affine transformation applied to the cropped image patch. To supervise the output of traditional explicit neural networks, the current heatmap generation strategy yields 2D pixel arrays to reflect the spatial distribution of body joints, which means that the originally continuous heatmap representation needs to be discretized as shown

---

\*Corresponding author.

38th Conference on Neural Information Processing Systems (NeurIPS 2024).

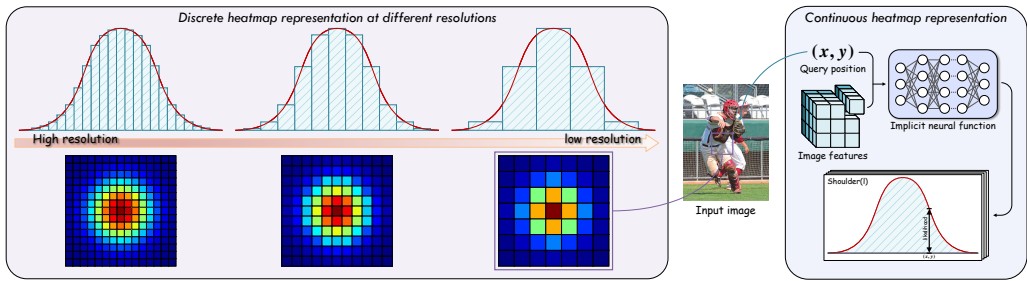

Figure 1: **Comparison of discrete and continuous heatmap representations.** In heatmap-based methods, the Gaussian function is discretized to satisfy the form of 2D pixel arrays. As the resolution decreases, the impact of quantization errors on positioning accuracy increases significantly. In contrast, NerPE can regress confidence scores at any position via implicit neural representation.

in Fig. 1. Specifically, the center of the Gaussian kernel is placed at body joints to calculate the confidence scores on the grid points as the ground-truth heatmaps. Although this discrete heatmap representation has achieved great success, it introduces quantization errors since body joints after affine transformation may fall anywhere within the image range rather than just at these fixed grid points. For the output of explicit neural networks, the information loss caused by spatial sampling is difficult to compensate through post-processing operations [31, 54]. With the reduction of the input resolution, this problem will be further exacerbated, which makes the performance of heatmap-based methods degraded even worse than some methods based on coordinate regression. Considering that the Gaussian function used for heatmap generation is inherently continuous, it is a natural idea to learn a continuous heatmap representation to replace fixed spatial sampling for HPE.

Implicit neural representations (INRs) are proposed to parameterize a variety of continuous signals [30, 8, 7, 47, 39, 46] in computer vision. Unlike explicit neural networks that output specific structures (*e.g.*, mesh, sequence), INRs map an index to its corresponding value, enabling not only continuous representations but also greater flexibility in use. Obviously, by combining INR and heatmap regression, the pose estimation model is able to learn continuous confidence scores for all body joints at any position. Compared to the discrete heatmap representation with a fixed resolution, the introduction of INR avoids the invariant spatial sampling caused by discretization during training and can yield the predicted heatmaps at arbitrary resolution during testing.

In this paper, to improve the performance of heatmap-based HPE, we abandon the main culprit of quantization errors, namely discrete heatmap representation. Instead, we propose NerPE to achieve continuous heatmap regression through a novel neural representation. Specifically, we generalize the upsampling factor in discrete sub-pixel convolution [38] to infinity to obtain a continuous upsampling function with respect to 2D coordinate. NerPE learns its continuous heatmap representation at a series of queried positions that are randomly sampled within the image range. For any position, the target likelihood at it is calculated from the target 2D pose and its absolute coordinate, and the estimated likelihood at it is inferred from the local feature vector and its relative coordinate. Since confidence scores are learned at all positions within the image range during training, our method achieves better performance than discrete heatmap-based methods that focus only on fixed grid points. As the resolution of the input image decreases, the superiority of continuous heatmap representation over existing methods becomes more prominent.

Limited by the explicit neural representation of existing methods, the heatmap resolution for a given image cannot be changed once the network structure is determined. In contrast, another benefit that INR brings to heatmap regression is that the heatmap resolution is no longer correlated with the image resolution. This means that our method can flexibly generate the predicted heatmaps with different resolutions based on accuracy requirements without retraining. To speed up the inference when high-resolution heatmaps are required, we design a progressive coordinate decoding method. Thanks to the decoupling of INR and spatial resolution, NerPE enables high-precision localization without the need to calculate the complete heatmaps. Specifically, low-resolution heatmaps are first output to determine the approximate locations of body joints. Subsequently, the area near the maximal activation is further retrieved in an iterative manner. Notably, our method can be easily integrated into most heatmap-based methods. The contributions are summarized as follows:

- We propose NerPE to avoid the introduction of quantization errors during training and to output the predicted heatmaps at arbitrary resolution during inference. To our knowledge, we are the first to apply implicit neural representations to human pose estimation.

- We design a progressive coordinate decoding method to derive the coordinates of body joints from continuous heatmap representation, in which our method achieves high-precision localization with low computational cost through coarse-to-fine retrieval.

- We conduct extensive experiments on three pose estimation benchmarks: COCO [25], MPII [1], and CrowdPose [21]. The results show that NerPE significantly enhances existing heatmap-based methods and obtains superior performance on low-resolution input images.

## 2 Related Work

### 2.1 Human Pose Estimation

2D human pose estimation (HPE) aims to locate a series of anatomical keypoints to represent the human pose in the input image. Currently, coordinate regression [3, 22, 28] and heatmap regression [17, 45, 12] are the two main pose estimation paradigms, both of which have received widespread attention. Coordinate regression, also known as direct regression, relies on deep neural networks to explore the mapping between the input image and the coordinates of body joints. The high degree of nonlinearity is difficult for optimization, which makes coordinate regression not perform well enough. As for heatmap regression, HPE is converted into a combination of multi-label keypoint classification and coordinate decoding post-processing. Benefiting from dense prediction, heatmap-based methods not only easily exploit visual cues around the target position, but also take the ambiguity of keypoint localization into account. This is why heatmap-based methods are generally superior to those based on coordinate regression. Although heatmap-based methods have made steady progress, quantization errors are still a troubling problem, which is what our work is dedicated to solving.

### 2.2 Discrete Heatmap Regression

To be clear at first, the cause of quantization errors is discretization, not the introduction of heatmap representations this behavior itself. As far as we know, existing heatmap-based methods [48, 42, 24, 52] all belong to discrete heatmap regression, using 2D pixel arrays to describe the spatial distribution of body joints. There are a few works that attempt to compensate for the damage to performance caused by quantization errors. In [31], the coordinates of body joints are empirically determined as the position of moving a 0.25 pixel from the maximal to second maximal activation. DARK [54] implements Taylor series approximation for heatmap activation to locate body joints based on distribution information. Although these post-processing operations achieve sub-pixel localization precision, subjective assumptions make the inference not so reliable, especially in low-resolution cases. Instead of remedying the shortcomings of discrete heatmap representation, we propose continuous heatmap regression to preserve the continuity of ground-truth heatmaps.

### 2.3 Implicit Neural Representation

In response to the need to model continuous signals, implicit neural representations (INRs) aim to learn a neural function that predicts the corresponding value according to a given index. Encouraged by the success in 3D reconstruction [14, 32, 4] and generation [6, 33, 49], INR has been extended to some other tasks, including super-resolution [8, 10, 13], image generation [40, 5, 2], and video compression [7, 15, 27]. In the paper, to avoid the damage of discretization to heatmap regression, we approximate the continuous Gaussian function by multi-layer perceptrons (MLPs). Given a coordinate within the image range, NerPE is designed to regress the confidence scores of all body joints at that position, in which the heatmap representation is free from the constraints of 2D pixel arrays [11, 51, 50]. As a result, NerPE has the ability to output the predicted heatmaps at arbitrary resolution during inference. In terms of eliminating quantization errors, offset prediction [34, 19] has the same goal as our work, and it aims to perform unbiased estimation for the coordinates of body joints. However, the performance of offset prediction is still affected by the resolution of the distance field. There is complementarity between the two techniques in HPE: offset prediction helps achieve more advanced decoding, and INR makes the distance field continuous. In order to highlight the superiority of INR, we only regress confidence scores in this paper.

# 3 Proposed Method

In this section, we first discuss the hazards of discrete heatmap representation and the limitations of existing post-processing operations. Then, we introduce a novel **Ne**ural **r**epresentation for human **P**ose **E**stimation (NerPE) and illustrate the superiority of continuous heatmap regression.

## 3.1 Preliminary of Heatmap Representation

Given an input image $I \in \mathbb{R}^{3 \times H_I \times W_I}$, heatmap-based models output 2D pixel arrays $H$ of size $K \times \frac{H_I}{4} \times \frac{W_I}{4}$ to reflect the spatial distribution of $K$ body joints. To conform the output form of explicit neural networks, the continuous Gaussian function is discretized in the heatmap generation process, which leads to the introduction of quantization errors and the reduction of positioning accuracy. Currently, there are two post-processing operations [31, 54] that are widely used to mitigate the negative effects of discretization.

The standard coordinate decoding method [31] is designed entirely according to experience. Based on the analysis of the model's performance, the position $p$ at which the maximal activation $H(c_m)$ moves a 0.25 pixel towards the second maximum $H(c_s)$ is taken as the final coordinate:

$$p = c_m + 0.25 \cdot \frac{c_s - c_m}{\|c_s - c_m\|_2}. \tag{1}$$

The distribution-aware decoding method [54] assumes that the predicted heatmaps still conform to the Gaussian distribution. The Taylor series expansion to the quadratic term is implemented at the position $c_m$ of the maximal activation as:

$$H(\mu) = H(c_m) + H'(c_m)(\mu - c_m) + \frac{1}{2}(\mu - c_m)^T H''(c_m)(\mu - c_m), \tag{2}$$

where $H'(\cdot)$ and $H''(\cdot)$ respectively denote the first and second order derivatives of the Gaussian distribution close to the predicted heatmaps. The position $p$ of each body joint is determined by the Gaussian mean $\mu = c_m - (H''(c_m))^{-1} H'(c_m)$.

To achieve sub-pixel localization precision against quantization errors, the above post-processing operations impose subjective assumptions in coordinate decoding. Compared with the limited improvements brought by the above passive compensations, the most direct way to solve quantization errors is to abandon the explicit neural network and learn a continuous heatmap representation.

## 3.2 Reformulation in a Continuous Form

From a structural and functional perspective, general pose estimation models consist of an encoder and a decoder (also called "head" in some works). To reduce the computational cost and increase the receptive field, the image features are downsampled in the encoder. Correspondingly, the decoder increases the output resolution by including upsampling layers such as interpolation, deconvolution or sub-pixel convolution [38]. However, once the network structure is determined, the decoder in existing methods makes the model only output the predicted heatmaps with a fixed resolution. Starting from sub-pixel convolution, we derive its continuous version via implicit neural representation (INR), known as a continuous upsampling function.

Using sub-pixel convolution as the final layer, the image features $Z$ of size $C \times H_Z \times W_Z$ are mapped into higher resolution heatmaps $H$ of size $K \times rH_Z \times rW_Z$ with an upscaling factor $r$:

$$H = \mathcal{PS}(W * Z + b), \tag{3}$$

where $\mathcal{PS}$ denotes a PixelShuffle operation used to reshape each element along the feature dimension into small 2D pixel arrays (called a cell) corresponding to body joints. According to the spatial prior, the weight $W$ and bias $b$ in Eq. (3) are further expressed as:

$$W = Concat(\begin{bmatrix} W_{1,1} & \cdots & W_{1,r} \\ \vdots & \ddots & \vdots \\ W_{r,1} & \cdots & W_{r,r} \end{bmatrix}), \quad b = Concat(\begin{bmatrix} b_{1,1} & \cdots & b_{1,r} \\ \vdots & \ddots & \vdots \\ b_{r,1} & \cdots & b_{r,r} \end{bmatrix}), \tag{4}$$

where the order of concatenation corresponds to the order of expansion in the PixelShuffle operation. These sub-weights $W_{i,j}$ and sub-biases $b_{i,j}$ can be viewed as the outputs of functions $f_W(i, j)$ and

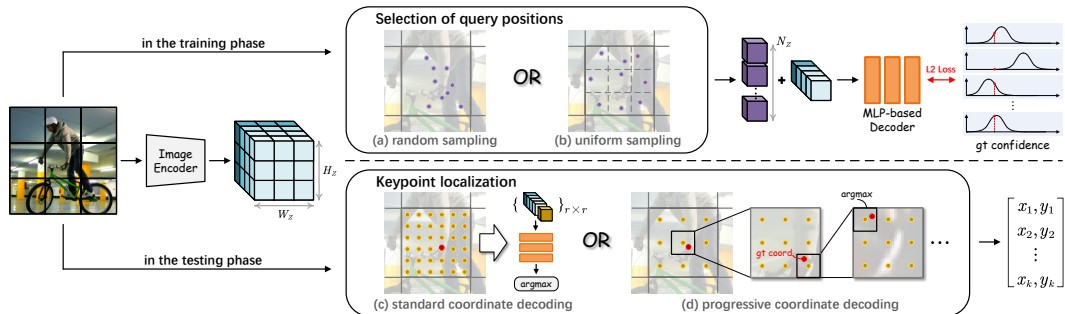

Figure 2: **Overview of NerPE.** The network structure consists of a general image encoder and an MLP-based decoder. During training, we use random or uniform sampling to pick queried positions, and calculate their confidence scores via continuous heatmap generation. During testing, we can obtain the predicted heatmaps at arbitrary resolution by standard and progressive coordinate decoding.

$f_b(i, j)$ with a 2D index as argument. We use $z^*$ to refer to each element in the image features $Z$, called a local feature vector. The confidence scores at $(i, j)$ relative to $z^*$ is given by:

$$H_{z^*}(i, j) = f_W(i, j) * z^* + f_b(i, j) = f_\theta(z^*, (i, j)). \tag{5}$$

In order to ensure the consistency in description of sub-pixel convolution with different upscaling factors, we normalize the 2D index array composed of integers in $\{1, 2, \ldots, r\}$ into $[-1, 1]$:

$$\begin{bmatrix} i' \\ j' \\ 1 \end{bmatrix} = \begin{bmatrix} \frac{2}{r} & 0 & 0 \\ 0 & \frac{2}{r} & 0 \\ 0 & 0 & 1 \end{bmatrix} \begin{bmatrix} 1 & 0 & -\frac{r+1}{2} \\ 0 & 1 & -\frac{r+1}{2} \\ 0 & 0 & 1 \end{bmatrix} \begin{bmatrix} i \\ j \\ 1 \end{bmatrix}. \tag{6}$$

As $r$ increases, the spatial sampling of $[-1, 1] \times [-1, 1]$ becomes denser. When $r$ approaches $\infty$, the discrete grid points become a continuous region covering all resolutions. Therefore, we obtain the expression for the continuous version of sub-pixel convolution $H_{z^*}(c_{rel}) = f_\theta(z^*, c_{rel})$, and further replace the relative coordinate $c_{rel}$ with the absolute coordinate $c_{abs}$ to get:

$$H(c_{abs}) = f_\theta \left( z^*, \frac{c_{abs} - c_{z^*}}{s_{cell}/2} \right), \tag{7}$$

where $c_{z^*}$ is the center coordinate of cell $H_{z^*}$, and $s_{cell}$ is the window size of cell $H_{z^*}$. Starting from sub-pixel convolution, the conclusion drawn is similar to the local implicit image function in [8] except for the additional normalization step. In terms of network architecture, we only replace the last few layers in existing heatmap-based models with our continuous upsampling function, which fundamentally solves the problem of discretization in training.

### 3.3 Learning Continuous Heatmap Regression

Obviously, the discrete heatmap representation is a subcase of the proposed NerPE, where only several fixed positions (*i.e.*, grid points) within the image range are trained and predicted. Thanks to the decoupling of heatmap representation and spatial resolution, NerPE can comprehensively fit Gaussian or Laplace functions. The design of our method involves three aspects: continuous heatmap generation, uniform position sampling, and progressive coordinate decoding.

**Continuous heatmap generation.** Since the output of the network is no longer 2D pixel arrays like existing methods, we need to calculate the ground-truth confidence scores for those positions being queried. In our method, the continuous heatmap representations based on Gaussian and Laplace distributions respectively are expressed relative to $c_{abs}$ as:

$$H_{gau}^{gt}(c_{abs}) = e^{-\frac{(c_{abs} - c_{gt})^2}{2\sigma^2}} \quad \text{and} \quad H_{lap}^{gt}(c_{abs}) = e^{-\frac{|c_{abs} - c_{gt}|}{b}}, \tag{8}$$

where $c_{gt}$ is the ground-truth coordinates of body joints. Since the queried position can be anywhere in the image range, the ground-truth heatmaps are continuous in space and confidence. The loss function of continuous heatmap regression is:

$$\mathcal{L} = \|H(c_{abs}) - H^{gt}(c_{abs})\|_2^2. \tag{9}$$

**Uniform position sampling.** For heatmap learning based on INR, the selection of queried positions for training is critical to the model performance. On the one hand, in order to fully take care of the entire heatmap plane, the sampling of queried positions during training should be evenly distributed over the image range. Thus, we divide each cell uniformly into $\sqrt{N_Z} \times \sqrt{N_Z}$ regions and perform random sampling within them, as shown in Fig. 2(b). In each training sample, $N_Z \times H_Z \times W_Z$ positions are picked out and their ground-truth confidence scores are calculated for supervision.

On the other hand, INR acts as a parameterized continuous function, which expects that spatially close positions should have similar confidence scores. Inside each cell, this is easy to implement for deep neural networks. However, at the junctions between cells, the difference of local feature vectors $z^*$ and the mutation of relative coordinates $c_{rel}$ make the network tend to produce discontinuous predictions. To solve this problem, we adopt the local ensemble in [8] to perform bilinear interpolation, where the sampling range is expanded to achieve overlapping.

**Progressive coordinate decoding.** The design of heatmap representation has been well discussed above, and more importantly, it ultimately serves keypoint localization. In order to decode the coordinates of body joints through *argmax*, a straightforward way is to arrange the candidate points within the cells at a specific density, so that NerPE can output the predicted heatmaps with the corresponding upsampling factor $r \geq 1$:

$$H_{cell}^{(n)} = f_\theta(z^{(n)}, C_{r \times r}), \tag{10}$$

where $H_{cell}^{(n)}$ is the distribution of body joints in the $n$-th cell of the input image, and $C_{r \times r}$ is a relative coordinate matrix to indicate the candidate points. When the heatmap resolution is high enough, *argmax* is sufficient to deal with quantization errors without extra post-processing operations.

In existing methods, the likelihood of body joints at each position is derived simultaneously by an explicit neural network, which means there is no choice but to output the entire predicted heatmaps. In contrast, NerPE can estimate confidence scores based on 2D coordinates in a serial manner thanks to the decoupling of INR from spatial resolution. We propose progressive coordinate decoding to reduce the computational cost by evading the complete generation of predicted heatmaps when the heatmap resolution is high, as shown in Fig. 2(d). Specifically, we first generate the low-resolution heatmaps to estimate the approximate locations of body joints. Then, the area near the maximal activation is iteratively sub-divided for coarse-to-fine retrieval. This decoding method only needs to calculate additional $t \times K \times (r + p_o) \times (r + p_o)$ positions to achieve an equivalent resolution of $r^t H_Z \times r^t W_Z$, where $t$ is the number of iterations and $p_o$ is the number of pixels overlapped.

## 4 Experiments

In this section, we compare the performance of NerPE and discrete heatmap-based methods on input images of different resolutions. The experimental results show the superiority of continuous heatmap regression and the hazards of discrete heatmap representation, especially in the case of low resolution.

### 4.1 Implementation Details

**Network architecture.** We adopt ResNet [16], HRNet [42] or TokenPose [24] as the backbone network, and resize the extracted image features to 8 × 8. The focus of this work is to provide a new perspective on heatmap regression through implicit neural representation (INR) rather than pursuing extreme performance. For clarity, we use a pure MLP structure as a decoder to implement continuous heatmap regression. Queried position embeddings, derived from 2D coordinates and their sinusoidal encodings, are concatenated with local feature vectors and then fed into the decoder. In each cell corresponding to a local feature vector, we collect $N_Z = 64$ queried positions as training subjects. For continuous heatmap generation, $\sigma$ in the Gaussian function is set to 0.06, which is close in proportion to the discrete heatmap representation. When not explicitly stated, the size of the predicted heatmaps is 256 × 256 by default. NerPE no longer needs those empirical operations to align flipped heatmaps [48] and shift decoded coordinates [31].

**Optimization.** In the main experimental results, the training settings of NerPE is consistent with the comparison methods [48, 42, 24] based on discrete heatmap regression. We use the Adam optimizer [18] for training, in which the learning rate is initialized to $1e-3$ and decreased to $1e-4$ and $1e-5$. The data augmentation used includes random rotation, random scale, image flipping, and half body cropping. All our experiments are conducted on an open-source machine learning, PyTorch [35].

Table 1: **Comparisons on the COCO validation set.** We report the performance of existing discrete methods and continuous NerPE at different input resolutions. OR/IR: the ratio of output resolution to input resolution. SimBa: SimpleBaseline. The best results are marked in **bold**.

| | Input size | Method | OR/IR | Params | AP | AR |
|---|---|---|---|---|---|---|
| ResNet-50 [16] | $64 \times 64$ | SimBa [48] | 1/4 | 34.0M | 34.4 | 43.7 |
| | | SimCC [23] | 3/1 | 24.7M | 39.3 | 48.4 |
| | | Ours | 4/1 | 29.6M | **40.8** (↑1.5) | **49.5** (↑1.1) |
| | $128 \times 128$ | SimBa [48] | 1/4 | 34.0M | 60.3 | 67.6 |
| | | SimCC [23] | 3/1 | 25.0M | 62.6 | 69.5 |
| | | Ours | 2/1 | 29.0M | **63.3** (↑0.7) | **70.1** (↑0.6) |
| | $256 \times 192$ | SimBa [48] | 1/4 | 34.0M | 70.4 | 76.3 |
| | | SimCC [23] | 2/1 | 25.7M | 70.8 | 76.8 |
| | | Ours | 1/1 | 28.4M | **71.0** (↑0.2) | **77.0** (↑0.2) |
| HRNet-W48 [42] | $64 \times 64$ | HRNet [42] | 1/4 | 63.6M | 48.5 | 57.8 |
| | | SimCC [23] | 3/1 | 63.7M | 59.7 | 67.5 |
| | | Ours | 4/1 | 63.9M | **62.5** (↑2.8) | **70.0** (↑2.5) |
| | $128 \times 128$ | HRNet [42] | 1/4 | 63.6M | 68.9 | 75.3 |
| | | SimCC [23] | 2/1 | 64.1M | 72.0 | 77.9 |
| | | Ours | 2/1 | 64.4M | **73.1** (↑1.1) | **78.8** (↑0.9) |
| | $256 \times 192$ | HRNet [42] | 1/4 | 63.6M | 75.1 | 80.4 |
| | | SimCC [23] | 2/1 | 66.3M | 75.9 | 81.2 |
| | | Ours | 1/1 | 65.0M | **76.1** (↑0.2) | **81.3** (↑0.1) |
| TokenPose-S [24] | $64 \times 64$ | DARK [54] | 1/4 | 4.9M | 57.1 | 64.8 |
| | | SimCC [23] | — | 4.9M | 62.8 | 70.1 |
| | | Ours | 4/1 | 5.4M | **64.4** (↑1.6) | **71.6** (↑1.5) |
| | $128 \times 128$ | DARK [54] | 1/4 | 5.2M | 65.4 | 71.6 |
| | | SimCC [23] | — | 5.1M | 70.4 | 76.4 |
| | | Ours | 2/1 | 5.5M | **71.8** (↑1.4) | **77.7** (↑1.3) |
| | $256 \times 192$ | DARK [54] | 1/4 | 6.6M | 72.5 | 78.0 |
| | | SimCC [23] | — | 5.5M | 73.6 | 78.9 |
| | | Ours | 1/1 | 6.5M | **73.9** (↑0.3) | **79.1** (↑0.2) |

Table 2: **Comparisons on the COCO test-dev set.** † indicates the ground-truth confidence scores of body joints calculated by the Laplace function in continuous heatmap generation.

| Method | Backbone | Input size | Params | AP | $AP_{50}$ | $AP_{75}$ | $AP_M$ | $AP_L$ | AR |
|---|---|---|---|---|---|---|---|---|---|
| SimBa [48] | ResNet-152 | $384 \times 288$ | 68.6M | 73.7 | 91.9 | 81.1 | 70.3 | 80.0 | 79.0 |
| HRNet [42] | HRNet-W48 | $384 \times 288$ | 63.6M | 75.5 | 92.5 | 83.3 | 71.9 | 81.5 | 80.5 |
| TokenPose [24] | HRNet-W48 | $256 \times 192$ | 27.5M | 75.1 | 92.1 | 82.5 | 71.7 | 81.1 | 80.2 |
| SimCC [23] | HRNet-W48 | $384 \times 288$ | — | 76.0 | 92.4 | 83.5 | 72.5 | 81.9 | 81.1 |
| Ours† | HRNet-W48 | $256 \times 192$ | 65.0M | 75.6 | 92.5 | 83.4 | 72.0 | 81.6 | 80.6 |
| Ours | HRNet-W48 | $384 \times 288$ | 65.0M | **76.2** | **92.6** | **83.6** | **72.8** | **82.0** | **81.2** |

## 4.2  Main Experimental results

**Evaluation on COCO.** To evaluate the value of continuous heatmap representation for human pose estimation (HPE), we perform NerPE with three backbones [16, 42, 24] at three input resolutions on the COCO validation set, as shown in Table 1. The experimental results show that our method achieves better performance, and the superiority over discrete heatmap-based methods increases as the input resolution decreases. Thanks to INR's modeling of continuous signals, NerPE can still output fine and smooth heatmaps of body joints even with a low-resolution image as input. In contrast, the discrete heatmap representation resorts to manually designed decoding methods [31, 54] to achieve sub-pixel accuracy, which suffers from significant performance degradation when quantization errors are too large. The comparison results on the COCO test-dev set are given in Table 2. These suggest that the accuracy and flexibility of the model are greatly improved by only replacing the last few layers in the existing heatmap-based methods with the INR-based MLP.

Table 3: **Comparisons on the MPII dataset.** The input resolution is $128 \times 128$ and the backbone is HRNet-W32. As a more stringent metric, PCKh@0.1 has higher requirements for localization.

| Method | Hea | Sho | Elb | Wri | Hip | Kne | Ank | PCKh@0.5 | PCKh@0.1 |
|---|---|---|---|---|---|---|---|---|---|
| HRNet [42] | 96.6 | 94.7 | 87.3 | 81.7 | 86.4 | 82.3 | 78.0 | 87.3 | 26.5 |
| DARK [54] | 96.6 | 94.5 | **87.7** | **82.2** | 87.2 | **82.8** | 78.4 | 87.6 | 29.6 |
| SimCC [23] | 96.6 | 94.6 | 87.5 | 81.3 | 86.8 | 82.5 | 78.2 | 87.4 | 32.6 |
| Ours† | 96.6 | 94.7 | 87.6 | 81.7 | 87.5 | 82.7 | 78.4 | 87.6 | 33.9 |
| Ours | **96.7** | **94.8** | **87.7** | 81.7 | **87.6** | **82.8** | **78.5** | **87.7** | **34.6** |

Table 4: **Comparisons on the CrowdPose dataset.** For the same backbone HRNet-W32, the impact of heatmap representation is given in the standard ($256 \times 192$) and low-resolution ($64 \times 64$) cases.

| Input size | Method | Continuity | AP | $AP_{50}$ | $AP_{75}$ | $AP_E$ | $AP_M$ | $AP_H$ |
|---|---|---|---|---|---|---|---|---|
| | HRNet [42] | $\times$ | 42.4 | 69.6 | 45.5 | 51.2 | 43.1 | 31.8 |
| $64 \times 64$ | SimCC [23] | $\times$ | 46.5 | 70.9 | 50.0 | 56.0 | 47.5 | 34.7 |
| | Ours | $\checkmark$ | **47.4** | **71.3** | **51.6** | **56.6** | **48.3** | **35.5** |
| | HRNet [42] | $\times$ | 66.4 | 81.1 | 71.5 | 74.0 | 67.4 | 55.6 |
| $256 \times 192$ | SimCC [23] | $\times$ | 66.7 | **82.1** | 72.0 | 74.1 | 67.8 | 56.2 |
| | Ours | $\checkmark$ | **66.9** | **82.1** | **72.6** | **74.2** | **68.0** | **56.4** |

**Evaluation on MPII.** We compare our NerPE with representative discrete heatmap-based methods [42, 54, 23] on the MPII dataset, as shown in Table 3. At the input size of $128 \times 128$, our method achieves better performance based on the same backbone HRNet-W32. The higher scores obtained on PCKh@0.1 indicate that NerPE's positioning of body joints is closer to the ground truth.

**Evaluation on CrowdPose.** To evaluate the performance in crowded scenes, we test NerPE on the CrowdPose dataset (see Table 4), in which YoloV3 [36] is adopted as the human detector. At the input size of $256 \times 192$, our method achieves superior performance with 66.9 AP. Thanks to the learned continuous heatmap representation, NerPE delivers performance gains on $AP_{75}$, a more stringent metric. For the low-resolution case, the experimental results show that NerPE brings an improvement of 6.1 AP to HRNet and further expands its lead over discrete heatmap-based methods.

**Progressive coordinate decoding.** In the proposed NerPE, the heatmap resolution is not dependent on the input resolution with the help of INR. Therefore, our method can flexibly increase the heatmap resolution to reduce quantization errors during inference. The standard NerPE directly outputs the complete predicted heatmaps for keypoint localization, which will bring a large amount of calculation when the heatmap resolution is high. To solve this problem, we propose

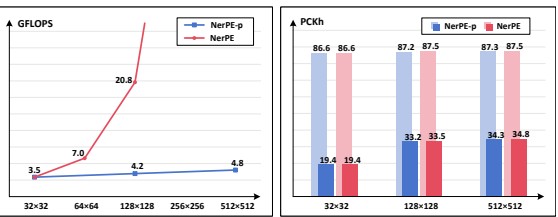

Figure 3: Comparison of computational cost (left) and accuracy (right) at different heatmap resolutions.

the progressive coordinate decoding method and denote the corresponding version as NerPE-p. Specifically, NerPE-p first yields coarse heatmaps of size $32 \times 32$. Then, we iteratively divide the area near the maximal activation into $4 \times 4$ and set $p_o$ to 2. Given inputs of size $128 \times 128$ from MPII, the comparison results of NerPE and NerPE-p are given in Fig. 3. As the heatmap resolution increases, the positioning accuracy is indeed improved but has a diminishing marginal effect. Compared with the standard NerPE, the use of our progressive coordinate decoding trades negligible performance degradation for a large reduction in computational cost.

## 4.3 Ablation Study

**Sample selection of INR.** In NerPE, we let the INR-related network learn the continuous heatmap representation covering the entire image through uniform position sampling. We perform ablations on the sampling modes (w/ and w/o uniform) and evaluate the impact of two hyper-parameters: the division of cells $H_Z \times W_Z$ (see Table 5) and the number of samples per cell $N_Z$ (see Table 6). The experimental results based on ResNet-50 at the input size of $128 \times 128$ show that the model

Table 5: **Ablation study on different divisions of cells**. The number of samples per cell is set to 64 on MPII (PCKh@0.5), using ResNet-50.

| Sampling | division of cells $H_Z \times W_Z$ | | |
|---|---|---|---|
| | $2 \times 2$ | $4 \times 4$ | $8 \times 8$ |
| w/o uniform | 75.28 | 80.83 | 82.54 |
| w/ uniform | 76.03 | 81.42 | 82.65 |

Table 6: **Ablation study on different number of samples per cell**. The division of cells is set to $8 \times 8$ on MPII (PCKh@0.5), using ResNet-50.

| Sampling | num_sample per cell $N_Z$ | | |
|---|---|---|---|
| | 4 | 16 | 64 |
| w/o uniform | 80.98 | 82.13 | 82.54 |
| w/ uniform | 81.45 | 82.49 | 82.65 |

Table 7: **Ablation study on scale parameters for continuous heatmap generation.** Experiments are performed on CrowdPose with input resolutions of $128 \times 128$. The backbone is HRNet-W32.

| Scale parameter | | AP | $AP_{50}$ | $AP_{75}$ | $AP_E$ | $AP_M$ | $AP_H$ |
|---|---|---|---|---|---|---|---|
| | 0.08 | 59.1 | 78.1 | 64.5 | 68.9 | 60.4 | 46.5 |
| $\sigma$ | 0.06 | 60.1 | 79.1 | 65.4 | 70.0 | 61.3 | 47.6 |
| | 0.04 | 59.5 | 79.1 | 64.6 | 69.1 | 60.6 | 47.2 |
| | 0.12 | 57.7 | 77.6 | 63.7 | 67.6 | 58.9 | 45.2 |
| $b$ | 0.09 | 58.8 | 78.5 | 64.6 | 68.4 | 60.1 | 46.5 |
| | 0.06 | 58.4 | 78.5 | 64.1 | 67.8 | 59.7 | 46.4 |

performance decreases when uniform position sampling is not used. For the division of cells, if it is too sparse (*e.g.*, $2 \times 2$), each local feature vector needs to be responsible for the prediction of a larger area, which is challenging for the network to fit the continuous heatmap representation. Considering both performance and computational cost, $8 \times 8$ is a better setting for NerPE. Furthermore, as the number of samples per cell increases, our method achieves better performance. The reason is that, on the one hand, querying more positions makes each gradient descent more robust, On the other hand, cells are decomposed into smaller regions to make the sampling more uniform.

**Study of heatmap generation.** In the Gaussian function, $\sigma$ is used as a hyper-parameter to control the scale of activation peaks. The difference between continuous and discrete heatmap representations has been discussed, as shown in Fig. 1. Due to the existence of discretization in existing methods, $\sigma$ is commonly set to an integer to facilitate the generation of the pixel-based Gaussian kernel (formally named standard biased encoding in [54]). In contrast, NerPE uses continuous coordinates rather than discrete indices to describe the heatmap plane, and the setting of $\sigma$ is more flexible under our continuous heatmap representation. The same conclusion goes for $b$ in the Laplacian function. We explore the influence of scale parameters $\sigma$ and $b$ on keypoint localization, as shown in Table 7. The proposed NerPE achieves better performance when $\sigma = 0.06$ and $b = 0.09$ respectively.

## 4.4 Visualization

In order to more intuitively show the superiority of continuous heatmap representation, we visualize the output of NerPE at different heatmap resolutions, as shown in Fig. 4. Thanks to the decoupling of INR from spatial resolution, NerPE can output the predicted heatmaps at arbitrary resolution without changing the structure and retraining the network. In addition, we visualize the results of using Gaussian and Laplace functions for supervision in continuous heatmap generation in Fig. 5.

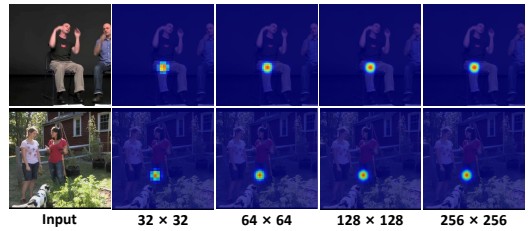

Figure 4: The predicted heatmap of knee(r) output by NerPE at different heatmap resolutions.

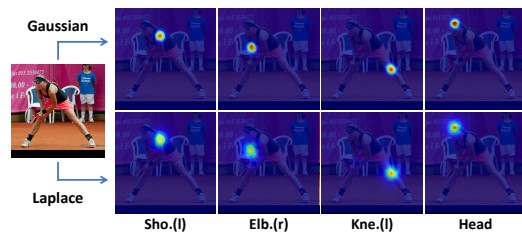

Figure 5: The output of NerPE supervised by different heatmap generation functions.

# 5 Conclusion

In this paper, to solve the quantization error issue plaguing heatmap regression, we propose an implicit neural representation method NerPE for 2D human pose estimation. According to the extracted image features, NerPE trains a simple MLP-based decoder to fit the Gaussian or Laplace functions at a series of queried positions, which makes the learned heatmap representation continuous in space and confidence. During inference, the decoupling from spatial resolution enables NerPE to output the predicted heatmaps at arbitrary resolution. As a result, our continuous heatmap regression achieves better performance than existing methods using discrete heatmap representation, especially in the case of low resolution. Last but not least, inspired by the flexibility of implicit neural representation, we design a progressive coordinate decoding method to speed up inference by avoiding the complete generation of predicted heatmaps when the desired heatmap resolution is quite high.

**Limitations and future work.** The goal of this work is to explore the feasibility of using implicit neural representations (INRs) to achieve continuous heatmap regression for 2D human pose estimation. To highlight the superiority of NerPE over discrete heatmap representation, our INR-based decoder is designed to be as simple as possible. In future work, we will conduct in-depth research on the network structure to better utilize the characteristics of INRs.

# Acknowledgements

This work was supported in part by the National Natural Science Foundation of China under Grant 62176125, Grant 61772272 and Grant 62406143, and in part by the Natural Science Foundation of Jiangsu Province under Grant BK20241468.

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

## A  Supplementary Implementation Details

As an implicit neural representation (INR) method, NerPE needs the correct coordinates to query the confidence scores of body joints during training and testing, but traditional coordinate transformation cannot meet our accuracy requirements. We give our data pre-processing and post-processing below.

### A.1  Coordinate Transformation in Data Pre-processing

In the proposed NerPE, image pixels are expected to be located at the center of their corresponding regions, which is not hold after affine transformation due to the implementation at the code level. In fact, for each region, the interpolation result of its upper left corner in the original image is used as its RGB value, as shown in Fig. A1. As a result, there is a spatial offset between the cropped image and its coordinate system $O_o$-$X_o$-$Y_o$. To solve this issue, we translate the 2D coordinates of body joints to an unbiased target coordinate system $O_u$-$X_u$-$Y_u$ to achieve alignment. Specifically, what needs to be done is to add 0.5 to the coordinates of body joints in $O_o$-$X_o$-$Y_o$.

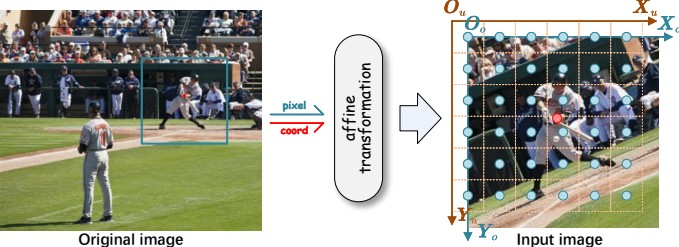

Figure A1: **Pre-processing of NerPE.** Due to differences in the affine transformations performed on pixels and coordinates, they are misaligned in the input image after standard data transformation. Therefore, we need to map the coordinates into $O_u$-$X_u$-$Y_u$ to achieve alignment.

### A.2  Coordinate Decoding in Data Post-processing

The schematic diagram of coordinate decoding in NerPE is shown in Fig. A2. To determine the positions of body joints during inference, first the *argmax* operation is performed on the predicted heatmaps $H$ to obtain a series of 0-based integral indices. Then, NerPE calculates the corresponding coordinates of these positions in $O_u$-$X_u$-$Y_u$. Finally, these coordinates are transferred to $O_o$-$X_o$-$Y_o$ for mapping back to the original image. The entire process is formulated as $p = (argmax(H) + 0.5) \cdot s - 0.5$, where $s$ represents the ratio of input resolution to heatmap resolution.

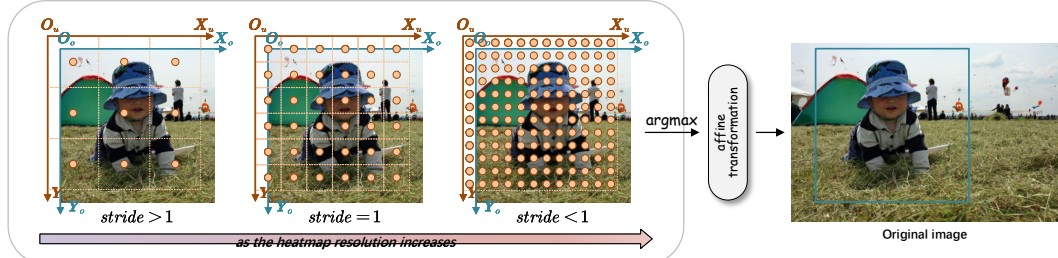

Figure A2: **Post-processing of NerPE.** Since the affine transformation is established between the cropped image and the original image, we need to convert the 0-based integral indices calculated by *argmax* into the coordinates in $O_o$-$X_o$-$Y_o$.

## B  Additional Visualization and Analysis

Here, we discuss in detail the local ensemble used in NerPE, and perform ablation on it as shown in Fig. A3. For the difference of local feature vectors $z^*$ and the mutation of relative coordinates

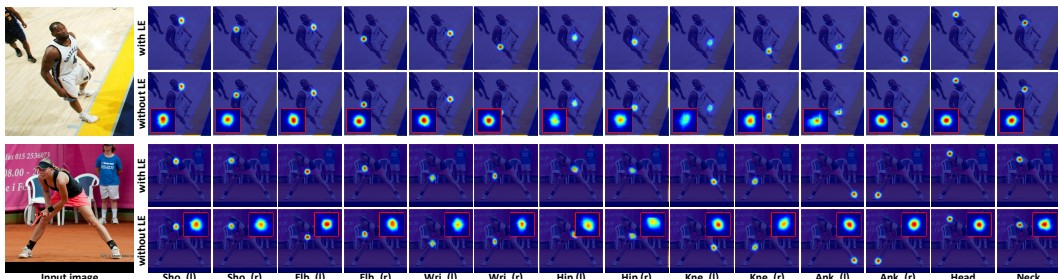

Figure A3: **Visualization of qualitative ablation on local ensemble.** The prediction of activation peaks is the key to heatmap-based pose estimation. When the activation peaks appear at the junction between cells, the confidence scores show obvious discontinuity without local ensemble (LE).

$c_{rel}$ at the junctions between cells, the local ensemble uses bilinear interpolation to ensure that the confidence scores output by the network is continuous. This process is formulated as:

$$H(c_{abs}) = \sum_{t \in \{00,01,10,11\}} \frac{S_t}{S} \cdot f_\theta \left( z_t^*, \frac{c_{abs} - c_{z_t^*}}{s_{cell}/2} \right),$$

where $z_t^*$ refers to the four local feature vectors that are closest to the queried position. The predicted confidence scores are weighted based on the surrounded areas $S_t$ and their sum $S = \sum_t S_t$. The use of local ensemble means that the sampling of queried positions is no longer limited to the interior of each cell, but the sampling range is expanded to twice the original to achieve overlapping. It can be found in Fig. A3 that the predicted heatmaps show discontinuity after removing the local ensemble.

