# OpenReview forum: "Continuous Heatmap Regression for Pose Estimation via Implicit Neural Representation"
_NeurIPS.cc/2024/Conference — NeurIPS 2024 poster_

### Official Review · Reviewer_rvLc · 2024-06-20

**Soundness:** 3
**Presentation:** 3
**Contribution:** 3
**Rating:** 6
**Confidence:** 4

**Summary:**

This paper introduces a new heatmap representation for 2D human pose estimation. Prior approaches use a quantized representation of heatmaps, where a confidence score is assigned to each pixel. In addition to being dependent on the image resolution, this representation does not match the usual ground truth coordinates, which are continuous when working in image crops.

This paper proposes an implicit neural representation to address this limitation of prior works. Specifically, given 2D coordinates (which do not necessarily correspond to the center of a pixel), the introduced neural network outputs a confidence score for each keypoint. At inference, the coordinates of each keypoint are recovered using the maximum confidence value. The main contributions of this paper are the following: (1) The authors introduce NerPE, a novel implicit representation of heatmaps for continuous human pose estimation; (2) A progressive coordinate decoding method is introduced to recover 2D coordinates from continuous heatmaps.

**Strengths:**

- The approach is novel; to my knowledge, no prior work has used implicit representations for heatmaps.

- The overall writing is good, and the paper reads nicely. The weakness of prior works addressed in this paper is clearly identified, and the proposed approach is well-justified.

- The proposed approach yields consistent improvement over the SOTA methods, and the ablation study gives a good idea of the impact of choice designs.

**Weaknesses:**

- The Related Work section lacks discussions and comparisons with other approaches attempting to address the quantization error in heatmaps, such as [42,a,b,c,d,e,f,g]. The authors say that "As far as we know, existing heatmap-based methods all belong to discrete heatmap regression" (L104), but for instance, [b] predicts a float location using an offset within the original heatmap. Similar approaches using PixelShuffle operation were proposed [f]. In general, the introduction presents the problem of heatmap quantization as if prior works ignored it, but plenty of works attempted to address this problem.

- I am unsure if having a continuous representation of heatmaps is useful. The ground truth is given in pixel coordinates on the full image. I agree that increasing the heatmap resolution until we reach the original image resolution is helpful, but is it useful to have a higher resolution than the original image? In the end, the ground truth is given in full pixel coordinates.

- Some information important for reproducibility is missing. For instance, what are the training and validation datasets and the number of epochs? Even if the authors say they used "standard settings" I believe this is not enough to reproduce the results.

- No information is provided regarding the material used for training and testing or the running times. This is particularly concerning as the method has a high computational cost. The authors propose an alternative decoding strategy that seems much less expensive, but comparing it with other methods and heatmap representations would be necessary.




[a] Bulat, A., Sanchez, E., & Tzimiropoulos, G. (2021). Subpixel heatmap regression for facial landmark localization. BMVC

[b] Lan, X., Hu, Q., & Cheng, J. (2021). Revisiting quantization error in face alignment. ICCV

[c] Yu, B., & Tao, D. (2021). Heatmap regression via randomized rounding. PAMI

[d] Papandreou, G., Zhu, T., Kanazawa, N., Toshev, A., Tompson, J., Bregler, C., & Murphy, K. (2017). Towards accurate multi-person pose estimation in the wild. CVPR

[e] Luvizon, D. C., Picard, D., & Tabia, H. (2018). 2d/3d pose estimation and action recognition using multitask deep learning. CVPR

[f] Wang, H., Liu, J., Tang, J., & Wu, G. (2023). Lightweight Super-Resolution Head for Human Pose Estimation. MM

[g] Li, J., Bian, S., Zeng, A., Wang, C., Pang, B., Liu, W., & Lu, C. (2021). Human pose regression with residual log-likelihood estimation. ICCV

**Questions:**

(1) "The meaning of the confidence scores output by current heatmap-based models has not been theoretically proven." (L43). What does this really mean? Is the NerPE representation "theoretically proven"?

(2) In Equation (5), to my understanding, $f_W(i,j)$ and $f_b(i,j)$ are floats, and $z^*$ would be of dimension $C$? But isn't $H_{z^*}$ supposed to be a float since it is a confidence value?

(3) "We are surprised to find that the conclusion drawn is similar to the local implicit image function in [8] except for the additional normalization step." (L179). Why is that surprising? Isn't the process a bit similar?

(4) Could we consider that the proposed method consists of doing sub-heatmaps until we reach a desired degree of precision?

(5) The model is trained to output Gaussian or Laplacian heatmaps. However, at inference, there is no guarantee that the confidence distribution follows the same distribution. I think that this is a good point, as this brings flexibility. However, couldn't that raise issues for finding the argmax? For instance, the model could potentially "hesitate" between 2 different places when placing the center of the heatmap (the predicted distribution would look like a mixture of Gaussians, for instance). Did you observe such cases in practice, and could other strategies be explored instead of choosing the argmax?


Side remark: Typo L15 "desgin"

**Limitations:**

Limitations of this work are briefly discussed in Section 5:

- The limited experiments on network architecture and index embedding are not a problem, as this work aimed to focus on the heatmap representation.

- The discussion on the necessity of predicting discrete heatmaps for computing an argmax is interesting, but it would have been more valuable if the authors had provided potential improvements and strategies on that point.

The potential negative societal impacts are not discussed but are rather limited. The authors could consider mentioning the potential misuse of pose estimation for intrusive surveillance or military applications. The environmental cost could also be discussed, especially since this method is more computationally expensive than prior works.

---

> ### Author Rebuttal · Authors · 2024-08-07
>
> We thank the reviewer for the thoughtful review. Below we address all the concerns.
>
> ***W1: The Related Work section lacks discussions and comparisons with other approaches attempting to address the quantization error ...***
>
> The quantization error problem caused by heatmap discretization is a well-known fact. In the Introduction, what we want to express is that the existing works are committed to overcoming quantization errors (as a result), while we hope to solve heatmap discretization (as a cause). Although some methods achieve sub-pixel positioning accuracy with the help of techniques such as offset prediction, their performance still depends on the quality of heatmap regression. We do not deny the achievements of current works on the quantization error problem, but provide a new perspective to solve it inspired by implicit neural representations. We will correct the text description to avoid this misunderstanding in the Introduction and distinguish our method from the works mentioned by the reviewer in Related Work.
>
> ***W2: I am unsure if having a continuous representation of heatmaps is useful. The ground truth is given in pixel ...***
>
> In the top-down paradigm, the image patches containing the person instances in the original image are scaled and rotated before being fed into the pose estimation network. As a result, the original pixel-level annotation becomes the floating-point ground truth after affine transformation. Therefore, in coordinate decoding, even if the heatmap resolution is equal to the input resolution, the coordinates calculated by argmax are still lossy.
>
> ***W3: Some information important for reproducibility is missing. For instance, what are the training and validation datasets ...***
>
> COCO, MPII, and CrowdPose are standard human pose estimation datasets with fixed settings in terms of train\val division, data augmentation, etc. The random rotation and scaling factors are [[-45°, 45°], [0.65, 1.35]] in COCO, [[-30°, 30°], [0.75, 1.25]] in MPII, and [[-45°, 45°], [0.65, 1.35]] in CrowdPose. "standard settings" means that the training process of NerPE is consistent with the corresponding baseline, where the learning rate is initialized to 0.001 and decayed twice by a factor of 0.1. The learning schedules with SimpleBaseline, HRNet, and TokenPose as the backbone are [90th, 120th, 140th] epoch, [170th, 200th, 210th] epoch, and [200th, 260th, 300th] epoch.
>
> ***W4: No information is provided regarding the material used for training and testing or the running times ...***
>
> We provide a comprehensive analysis of the efficiency and performance in **Tab. A3 of the attached PDF**, where we supplement GFLOPs and inference time to better compare NerPE with existing methods. As shown in Tab. A3, our progressive coordinate decoding effectively solves the shortcoming of implicit neural representations that generate a lot of computation when outputting high-resolution signals.
>
> ***Q1: "The meaning of the confidence scores output by current heatmap-based models has not been theoretically proven." ...***
>
> Existing methods regress heatmaps in the form of 2D pixel arrays. To generate discrete ground truth heatmaps, many influential works (such as HRNet, ViTPose) place a fixed Gaussian kernel on the grid points to which body joints belong. Therefore, the ground truth heatmaps have a series of fixed discrete values while the heatmaps predicted by the neural network have continuous values, which are not equivalent. In contrast, our generated ground truth heatmaps are continuous, so there is no such problem.
>
> ***Q2: In Equation (5), to my understanding,*** $f_{W}(i,j)$ ***and*** $f_{b}(i,j)$ ***are floats, and*** $z^{*}$ ***would be of dimension*** $C$ ***? But ...***
>
> In Eq. 5, $f_{W}(i,j)$ and $f_{b}(i,j)$ are rewritten from $W_{i,j}$ and $b_{i,j}$, where the mapping $f$ represents the selection in the weight libraries $W$ and $b$ according to an index, i.e., $f:(i,j)\mapsto W_{i,j},b_{i,j}$. The local feature vector $z^* \in \mathbb{R}^{C \times 1 \times 1}$ is a element in the image features $Z \in \mathbb{R}^{C \times H_{Z} \times W_{Z}}$, and $H_{z^*}$ denotes the cell in the heatmaps $H$ corresponding to $z^*$. So, $H_{z^*}$ is a float tensor instead of a float.
>
> ***Q3: "We are surprised to find that the conclusion ..." (L179). Why is that surprising? Isn't the process a bit similar?***
>
> In [Ref1], the local version of implicit neural representation is proposed based on intuition, and at least the persuasive derivation process is not provided in its paper. In our work, we derive a similar conclusion from sub-pixel convolution, which is exciting because we find a reasonable explanation for the local implicit neural representation.
>
> ***Q4: Could we consider that the proposed method consists of doing sub-heatmaps until we reach a desired degree of precision ?***
>
> Yes. Our method has two ways to reach the desired degree of precision. The standard NerPE directly outputs complete predicted heatmaps at the resolution corresponding to the required precision. The variant NerPE-p achieves equivalent resolution through coordinate decoding to iteratively refine the local parts of heatmaps, as the reviewer said.
>
> ***Q5: The model is trained to output Gaussian or Laplacian heatmaps. However, at inference, there is no guarantee that ...***
>
> As the reviewer said, in existing heatmap regression research, the distribution of predicted heatmaps is generally not guaranteed to conform to the Gaussian or Laplace distribution. The reason is that the loss function is the average of the differences of each heatmap pixel, so our method is not immune even if our heatmap representation is continuous. To solve this problem, it may be a simple and effective solution to add predictions of mean and variance to assist argmax.
>
> [Ref1] Learning continuous image representation with local implicit image function. CVPR, 2021.

---

> > ### Comment · Reviewer_rvLc · 2024-08-10
> >
> > Thanks a lot to the authors for this rebuttal, which addressed all my questions and concerns. I still think that this paper should be accepted.
> >
> > I think it is important to add the information from Table A3 in the final version; I expected the running times to be larger given the inference process.

---

> > > ### Author Response · Authors · 2024-08-11
> > >
> > > We thank the reviewer for the positive feedback. We are glad that the reviewer thinks our work is worthy of acceptance.
> > >
> > > > I think it is important to add the information from Table A3 in the final version; I expected the running times to be larger given the inference process.
> > >
> > > We will report the analysis of efficiency in our paper. There are two reasons why the inference time is faster than expected: (1) In both NerPE and NerPE-p, the bakcbone network is executed only once in the forward pass. (2) The INR-related part is implemented by a simple MLP, and queries for multiple positions are performed in parallel on the GPU. Therefore, the inference time of our method is close to that of our baseline.

---

### Official Review · Reviewer_KNNC · 2024-07-07

**Soundness:** 3
**Presentation:** 3
**Contribution:** 3
**Rating:** 5
**Confidence:** 5

**Summary:**

This paper proposes a new approach to predict continuous heatmap for keypoint localization. The proposed method adopts a MLP that receives position coordinates and corresponding feature and output the confidence score of this position. Then this method can query candidate points and select the point with maximum score as final results. Experiments on several keypoint localization benchmarks demonstrate the effectiveness of the proposed method.

**Strengths:**

1.	The idea of introducing implicit neural representation to perform keypoint localization is new and can addresses the quantization error of existing heatmap-based method.
2.	This paper is well-written and easy to understand.

**Weaknesses:**

My major concern is about the decoding process of the proposed method. It seems that this method should evaluate a lots of points to select the best results, which will introduce much inference computational cost. The introduced progressive coordinate decoding can reduce the computational cost but can avoidably increase the decoding time due to multi-round decoding process. This paper should give a detailed efficiency comparison of existed methods such as heatmap (SimpleBaseline, HRNet, SimCC, ViTPose), regression (RLE), including GFLOPs, inference time (ms), GPU memory consumption under the same setting (all above methods release their codes, so it is not impossible to conduct such comparisons). These comparisons are necessary because it seems that the advantages of the proposed method is not obvious.

**Questions:**

In Weakness

**Limitations:**

This paper discusses its limitation in Section 5.

---

> ### Author Rebuttal · Authors · 2024-08-07
>
> We thank the reviewer for the helpful review. Below we address all the concerns.
>
> ***Response to weaknesses:***
>
> ***Efficiency:*** We compare the efficiency and performance of NerPE and existing methods as shown in **Tab. A3 of the attached PDF**. Although implicit neural representations help us realize continuous representation, it has the disadvantage of a large amount of computation when generating high-resolution signals, just like its application in other computer vision fields. In Tab. A3, NerPE-p significantly reduces the GFLOPs to an acceptable level (from 73.6 to 9.5) by our progressive coordinate decoding that is designed according to the characteristics of human pose estimation.
>
> ***Performance:*** NerPE has a positive impact on the performance of the baseline and can be generalized to most heatmap-based methods with minor structural changes. We need to be clear that the quantization error problem worsens as the input resolution decreases. The introduction of implicit neural representations in NerPE significantly improves the performance of the baseline on low-resolution images (e.g., 64×64), as shown in Tabs. 1 & 4. In Tab. 3, the same conclusion is also illustrated by the large lead of NerPE in PCKh\@0.1.

---

### Official Review · Reviewer_k3b9 · 2024-07-12

**Soundness:** 3
**Presentation:** 4
**Contribution:** 3
**Rating:** 6
**Confidence:** 5

**Summary:**

The authors tackle 2D human pose estimation through a continuous heatmap representation. Specifically, instead of representing the heatmap as a grid of values, they use an implicit neural representation, where the coarse feature vector along with queried coordinates are fed to an MLP, which predicts the heatmap value for that requested position. This can in theory provide infinite resolution for the heatmap, allowing more accurate localization of its peak, compared to the usual discretized heatmaps that can lead to quantization errors due to the low resolution. The model is trained with randomized query points with Gaussian target. For test time, the authors propose a progressive refinement scheme where the highest-value area in the heatmap is successively queried at finer levels.
Experiments are performed on COCO, MPII and CrowdPose with improvements over baselines, especially for evaluations with stricter thresholds.

**Strengths:**

This is a creative and well-motivated approach. Whereas prior works often used heuristics such as moving the output point towards the second-highest heatmap pixel by a factor of 0.25 etc., this work is a more principled solution to producing arbitrary-resolution heatmaps.

The experiments are extensive, with three different backbones and three different datasets. The most convincing of these is the MPII experiment with the strict threshold, showing a clear improvement over comparable baselines. This is exactly the setting where one would expect to see the improvement. The other settings and dataset also see some quantitative improvement, which substantiates the claims of the paper.

There is little overhead in terms of parameter count and computational cost, and the technique can be applied to a wide range of approaches giving the potential for wide impact.

The writing has high quality and the structure is clear.

**Weaknesses:**

Some important alternative techniques from the literature are not discussed, and this impacts the conclusions.
First, soft-argmax (integral regression) [1] was introduced specifically to enable continuous output that is not limited by the coarse downsampled grid. This would be an important baseline decoding method to use here. (There have been several extensions, including [2])
Second, offset regression such as the short-range offsets learned in PersonLab [3] would be another simple formulation allowing continuous output not limited by the grid.

The statement around L305 is not correct, the radius of the Gaussian does not have to be an integer for the gridded heatmap. There is no connection between these two, it is neither easier nor more appropriate to use integer-valued standard deviations for a Gaussian on a grid (alternatively, clarification is needed).

[1] Sun et al., Integral Human Pose Regression, ECCV 2018
[2] Gu et al., Bias-Compensated Integral Regression for Human Pose Estimation, PAMI 2023
[3] Papandreou et al. PersonLab: Person Pose Estimation and Instance Segmentation with a Bottom-Up, Part-Based, Geometric Embedding Model, ECCV 2018

**Questions:**

Why is the proposed method not tested also on resolution 384x288? This would allow better comparability in Table 2. Further, why is the gap performance advantage significantly smaller on COCO test-dev compared to COCO val?

Were other alternatives considered for the implicit function, such as learning a distance field?
As for the progressive refinement strategy, wouldn't it be useful to base the exploration on the gradient of heatmap value wrt query location? This seems cheaply differentiable and could give a more direct path towards the peak.

Since the target is always a Gaussian, how does the network output at test time look? Are less certain predictions wider in distribution?

As the queries are given as coordinates to the MLP, would it be possible to query truncated joints outside the heatmap bounds?

**Limitations:**

The authors provide an adequate "Limitations" section in the paper.

---

> ### Author Rebuttal · Authors · 2024-08-07
>
> We thank the reviewer for the positive feedback. Below we address all the concerns.
>
> ***W1: Some important alternative techniques from the literature are not discussed, and this impacts the conclusions. First, ...***
>
> Here we analyze and compare our proposed method HerPE with the alternative techniques mentioned by the reviewer. As for Integral and its extension methods, from the perspective of supervision signals, they are essentially classified as coordinate regression, which is why they are not troubled by quantization errors. The original intention of these methods to construct a heatmap-like form is to narrow the gap between coordinate regression and heatmap regression, rather than to overcome the limitations of the coarse downsampled grid. In contrast, our work aims to propose a more advanced heatmap representation scheme to improve the performance of most heatmap-based methods. PersonLab consists of heatmap regression and offset prediction. Although PersonLab can achieve sub-pixel positioning with the help of offset prediction, its performance still depends on the quality of heatmap regression. When the output resolution of the heatmap and offset field is not high enough, the performance of PersonLab is also unsatisfactory. Therefore, our continuous heatmap representation also applies to PersonLab.
>
> ***W2: The statement around L305 is not correct, the radius of the Gaussian does not have to be an integer for the gridded heatmap ...***
>
> There is a mismatch between theory and practice. Most studies on heatmap-based human pose estimation only describe heatmap generation at a conceptual level using words or formulas, without discussing implementation in their papers. According to the official open source code, placing a fixed Gaussian kernel on the heatmap plane (i.e., integer Gaussian radius) is a standard operation in many influential works (such as SimpleBaseline, HRNet, HRFormer, ViTPose). To eliminate ambiguity, we will emphasize this detail in our paper and state that not all heatmap-based methods have this problem.
>
> ***Q1: Why is the proposed method not tested also on resolution 384x288? This would allow better comparability in Table 2. Further, ...***
>
> We report the results of NerPE on the COCO test-dev set with the input resolution of 384x288 as follows, to supplement Tab. 2. The reason for the different performance gap between NerPE and the comparison methods is that the input resolution is the same in Tab. 1 and different in Tab. 2.
>
> | Backbone | AP  | AP$_{50}$ | AP$_{75}$ | AP$_{M}$ | AP$_{L}$ | AR  |
> | --- | --- | --- | --- | --- | --- | --- |
> | HRNet-W48 | 76.2 | 92.6 | 83.6 | 72.8 | 82.0 | 81.2 |
>
> ***Q2: Were other alternatives considered for the implicit function, such as learning a distance field? As for the progressive refinement strategy, ...***
>
> There is no competition between our work and technologies such as distance fields. The purpose for us introducing implicit neural representations in human pose estimation is to achieve the transition from discrete to continuous. We use heatmap representation as the initial object to demonstrate the benefits of continuity, and the achievements can be generalized to other types of values ​​with 2D pixel arrays, also applicable to the distance field. In fact, the reviewer’s suggestion to use the gradient of heatmap value is to combine heatmap regression and offset prediction from the perspective of implicit neural representations.
>
> ***Q3: Since the target is always a Gaussian, how does the network output at test time look? Are less certain predictions wider in distribution ?***
>
> The trained NerPE is able to represent continuous heatmap distributions. The most direct way to obtain the coordinates of body joints is to query the confidence scores on the grid points according to the required resolution. Without retraining, visualization of the network output at different resolutions is given in **Fig. A2 of the attached PDF**.
>
> ***Q4: As the queries are given as coordinates to the MLP, would it be possible to query truncated joints outside the heatmap bounds ?***
>
> Unfortunately, in the current experimental setup, it is not possible to localize body joints that are outside the heatmap bounds due to differences in data distribution. If supervision information on truncated joints is provided for training, it is theoretically feasible to query whether there are body joints at a certain position outside the image range.

---

> > ### Comment · Reviewer_k3b9 · 2024-08-08
> >
> > Thank you for the detailed reply to my review.
> >
> > > The original intention of these methods to construct a heatmap-like form is to narrow the gap between coordinate regression and heatmap regression, rather than to overcome the limitations of the coarse downsampled grid.
> >
> > The issue of quantization errors (coarseness of the grid) is an explicit motivation for soft-argmax approaches [a] and [b], so there is considerable motivational overlap.
> >
> > [a] Nibali et al., Numerical Coordinate Regression with Convolutional Neural Networks. arxiv:1801.07372, 2018
> > [b] Sun et al., Integral Human Pose Regression. ECCV, 2018
> >
> > > Although PersonLab can achieve sub-pixel positioning with the help of offset prediction, its performance still depends on the quality of heatmap regression.
> >
> > This may be true, however the offsets do enable a continuous output without quantization errors (not bound to grid points).
> >
> > > When the output resolution of the heatmap and offset field is not high enough, the performance of PersonLab is also unsatisfactory.
> >
> > I would expect that NerPE's performance is similarly impacted if the underlying feature map has lower resolution (even though the implicit heatmap is continuous). Is it not so?
> >
> > Overall, I believe it would be important to discuss both of these prior lines of works in the paper. Even if they aren't doing the same thing, the motivation is similar enough.
> >
> > > We report the results of NerPE on the COCO test-dev set with the input resolution of 384x288 as follows
> >
> > Thank you, this result is convincing.

---

> > > ### Author Response · Authors · 2024-08-09
> > >
> > > Thank you for your kind reply!
> > >
> > > > Overall, I believe it would be important to discuss both of these prior lines of works in the paper. Even if they aren't doing the same thing, the motivation is similar enough.
> > >
> > > We will mention the soft-argmax approaches and PersonLab in Related Work to show the difference in solving the quantization error problem.
> > >
> > > > I would expect that NerPE's performance is similarly impacted if the underlying feature map has lower resolution (even though the implicit heatmap is continuous). Is it not so?
> > >
> > > For images with different resolutions and different backbone networks, we resize the encoded feature map $Z$ to 8×8 (L233), which means that each local feature vector $z^*$ is responsible for a 64-equally divided area on the 2D plane, independent of the resolution.
> > >
> > > If you have further questions, we are happy to discuss them.

---

> > > > ### Comment · Reviewer_k3b9 · 2024-08-12
> > > >
> > > > > For images with different resolutions and different backbone networks, we resize the encoded feature map to 8×8 (L233), which means that each local feature vector is responsible for a 64-equally divided area on the 2D plane, independent of the resolution.
> > > >
> > > > Based on the ablation in Table 5, NerPE's performance also depends on the grid resolution, so my point is that this aspect is not so different from the above-mentioned methods which can output continuous values but have worse performance on coarser grids.

---

> > > > > ### Author Response · Authors · 2024-08-13
> > > > >
> > > > > Thanks to the reviewer for further discussion on the concerns. First of all, we need to clarify again that the division of cells (i.e., grid resolution) in NerPE is a hyperparameter independent of the input resolution. We distinguish the division of cells and heatmap resolution from both theoretical and experimental aspects to show the differences between NerPE and the above-mentioned methods as follows:
> > > > >
> > > > > We are pleased that the reviewer agrees that the compared methods are still affected by heatmap resolution even if they output continuous values through additional operations. Due to information loss caused by heatmap discretization, the performance deteriorates as the resolution decreases. In contrast, the division of cells plays a different role than the heatmap resolution. For local implicit neural representations, if the division is too coarse, it is challenging for each local feature vector to be responsible for regressing a larger area in the image plane. If the division is too fine, the importance of the image information will overshadow the coordinate information because the confidence scores in each cell change marginally. To illustrate this, we add the case where the division of cells $H_Z \times W_Z$ is 64×64 based on Tab. 5, as shown in the following table. The results show that as $H_Z \times W_Z$ increases, the performance of NerPE first increases and then decreases, which is inconsistent with the trend of the impact of heatmap resolution.
> > > > >
> > > > > |     | 2×2 | 4×4 | 8×8 | 64×64 |
> > > > > | --- | --- | --- | --- | --- |
> > > > > | w\o uniform | 75.28 | 80.83 | 82.54 | 82.17 |
> > > > > | w\ uniform | 76.03 | 81.42 | 82.65 | 82.25 |
> > > > >
> > > > > In summary, although resolution is related to the performance of both NerPE and the above-mentioned methods, the principles and impacts are different. Furthermore, thanks to implicit neural representations, NerPE achieves decoupling from the output resolution, which can also be applied to techniques such as PersonLab to learn continuous distance fields.

---

### Official Review · Reviewer_xx7e · 2024-07-14

**Soundness:** 2
**Presentation:** 3
**Contribution:** 2
**Rating:** 5
**Confidence:** 5

**Summary:**

This paper mainly studies the quantization problem of discrete representation of heatmap, especially in the case of low resolution. It proposes to use a continuous Implicit Neural Representation (coordinate-input MLP) to coarse-to-fine query to generate heatmap at arbitrary resolution, which achieves better performance than the current common heatmap method at lower resolution.

**Strengths:**

- Heatmap quantization error has always been an important research topic, esp. at not high resolution.

- This paper is well-motivated and written in a simple and easy-to-understand way.

**Weaknesses:**

I will give the boardline first, and then want to see how the author responds to questions about method details, clarification of experimental settings, insufficient experiments, etc. before making a decision.

[Method]

- For training (Fig. 2), why not consider adding additional supervision in the area near the ground truth or even progressive coordinate decoding to refine the heatmap prediction near the ground truth? Will not this give better results?

- So does the final 2D coordinate input into the INR MLP use positional embedding (L312)?

- The bias of confidence estimation comes not only from discretization but also from heuristic confidence estimation [a]. From the gap (confidence) between AP (accuracy + confidence) and AR (accuracy) in Tab. 1, we could see that confidence calibration is still not good enough, and this limitation should be mentioned in the Sec. 5 Limitation & Future Work as well.

[Experiments]

- Does the backbone of this method use COCO/heatmap pre-trained or ImageNet checkpoint? Please describe in detail.

- I don't understand the meaning of the OR/IR column (Tab. 1). Why is HRNet's OR/IR constant at 1/4 and does not change with the input size?

- I don't seem to find GFLOPs, AP and AR results comparing different heatmap resolutions of the heatmap-based baseline method under a fixed input size? They may need to be added in Tab. 1.

- In addition to 256x192, it is also necessary to conduct experiments on a higher resolution 384x288 standard setting to test the applicability of the method's solving quantization error. It seems that the improvement of the method at high resolution is minor.

- How are the overhead savings and performance improvements on the newer heatmap-based method ViTPose [48]? Because ViTPose is powerful, and improving it will make this work more widely applicable.

- There are also some methods that use the heatmap + offset prediction method [b], which also needs to be discussed.

- If I understand correctly, I guess that progressive coordinate decoding may perform similarly to a fully generated heatmap in easier cases, and may only find local minima in more difficult cases (such as multi-peak, non-Gaussian non-uniform heatmaps). If this is the case, then splitting the test by difficulty (such as number of joints, human size, etc.) [c] may be more clear.

- Since the values ​​don't match, are the settings of Tabs. 5 & 6 ablation studies (82.65) different from Tab. 3 (87.7)?

- If the division in Tab. 5 is 16x16 or even higher resolution, the result should be improved further, though the cost gradually approaches the heatmap-based method? Could you add more reports in these regions (not necessarily done during the rebuttal)?

- The author mentioned that the difference in local feature vectors at the cell junction will make the predicted heatmap not smooth, so they proposed to use bilinear interpolation of local features (Ls202-207). Could we perform an ablation study to see the difference in results before and after and visualize the heatmap difference? I think we may also further consider the idea of ​​sparse convolution to fuse neighborhoods.

- It is recommended that the ablation study in Tab. 7 be performed more standardized on COCO 128x128 (not necessarily made during the rebuttal period) because of CrowdPose biases towards crowded situations. The 64x64 input resolution is too small and does not seem to have much application in practice.

- Lack of heatmap visualization and comparisons with other methods

References:

[a] On the calibration of human pose estimation. ICML, 2024.

[b] Towards accurate multi-person pose estimation in the wild. CVPR, 2017.

[c] Benchmarking and error diagnosis in multi-instance pose estimation. ICCV, 2017.

**Questions:**

Please see the weaknesses.

**Limitations:**

The authors state their limitations as 1. the decoder structure not studied in depth, 2. discrete heatmaps still required to generate and considered as not elegant. Besides, some limitations mentioned in the weaknesses are suggested to include as well.

---

> ### Author Rebuttal · Authors · 2024-08-07
>
> We thank the reviewer for the insightful comments. Below we address all the concerns.
>
> ***W1: For training (Fig. 2), why not consider adding additional supervision ...***
>
> Theoretically, additional supervision in the area near the ground truth can improve the performance of NerPE, but it also applies to discrete heatmap regression, which does not highlight the advantage of continuous representation. As for progressive coordinate decoding, it is only performed during inference.
>
> ***W2: So does the final 2D coordinate input into the INR MLP use positional embedding (L312) ?***
>
> We use a common high-frequency function $\gamma(x)=(\cdots,\sin (2^{i} \pi x),\cos(2^{i} \pi x),\cdots)$, to encode coordinates as sinusoidal embeddings. The relevant description will be added to the paper.
>
> ***W3: The bias of confidence estimation comes not only from discretization but also ...***
>
> Since NerPE is designed for quantization errors, the performance gains from confidence calibration over existing methods increase as the input resolution decreases. Our continuous representation does not conflict with the correction for heuristic confidence estimation. We will mention this in  Limitation & Future Work.
>
> ***W4: Does the backbone of this method use COCO/heatmap pre-trained or ImageNet checkpoint?***
>
> NerPE has the same backbone and loss function as its baseline. Therefore, the training process is consistent with its baseline, where the model is initialized with the pre-trained weights from ImageNet.
>
> ***W5: I don't understand the meaning of the OR/IR ...***
>
> OR is the output resolution and IR is the input resolution, which is explained in the caption of Tab. 1. Since HRNet belongs to explicit neural representation, once its network structure is determined, OR/IR becomes a constant.
>
> ***W6: I don't seem to find GFLOPs, AP and AR results ...***
>
> We report the efficiency and performance in **Tab. A3 of the attached PDF**. The high amount of computation is the disadvantage of implicit neural representations (INR) when generating high-resolution signals. The results show that we reduce the GFLOPs from 73.6 to 9.5 by progressive coordinate decoding.
>
> ***W7 & W8: It is also necessary to conduct experiments on a higher resolution ... & How are the overhead savings and performance improvements on ViTPose ?***
>
> Quantization errors caused by discretization decrease as the input resolution increases. Naturally, NerPE achieves greater improvements at low resolution than at high resolution. When the input resolution is 388×284, the accuracy of NerPE with HRNet and ViTPose as the backbone on the COCO test-dev set is as follows.
>
> | Backbone | AP  | AP$_{50}$ | AP$_{75}$ | AR  |
> | --- | --- | --- | --- | --- |
> | HRNet-W48 | 76.2 | 92.6 | 83.6 | 81.2 |
> | ViTPose-B | 76.8 | 92.6 | 84.3 | 81.8 |
>
> ***W9: There are also some methods that use the heatmap + offset prediction method ...***
>
> The heatmap + offset prediction method can indeed achieve sub-pixel positioning accuracy but it still depends on the quality of heatmap regression. When the resolution of heatmaps and offset fields is not high enough, its performance is also unsatisfactory. Therefore, our method not only does not form a competitive relationship with it but can even be combined with it.
>
> ***W10: If I understand correctly, I guess that progressive coordinate decoding ...***
>
> I agree. The intention of progressive coordinate decoding is to find a simple way to accelerate inference at the cost of slight performance degradation, as reported in Fig. 3. Indeed, thanks to decoupling from heatmap resolution, there are some better potential implementations for coordinate decoding, as the reviewer mentioned.
>
> ***W11: Since the values ​​don't match, are the settings of Tabs. 5 & 6 ablation studies (82.65) different from Tab. 3 (87.7)?***
>
> The backbone of NerPE in Tab.3 is HRNet-W32 and in Tabs. 5 & 6 is ResNet-50, which is given in the description of the corresponding experiments but may not be eye-catching enough. We will highlight it in the caption.
>
> ***W12: If the division in Tab. 5 is 16x16 or even higher resolution, the result should be ...***
>
> The results to divide the cells into 16×16 and 64×64 are shown below. Combined with Tab. 3, we find that 16×16 is indeed a better choice than 8×8. When it continues to increase to 64×64, the positioning accuracy decreases. This is because the over-fine division causes the degradation of NerPE. Since the confidence score in each region changes marginally, INR is more focused on image features than relative coordinates.
>
> |     | 16×16 | 64×64 |
> | --- | --- | --- |
> | w\o uniform | 83.13 | 82.17 |
> | uniform | 83.07 | 82.25 |
>
> ***W13: The author mentioned that the difference in local feature vectors at the cell junction will ...***
>
> The local ensemble is quantitatively proven to be effective for local INR in its original paper. In **Fig. A1 of the attached PDF**, we supplement the qualitative ablation of local resemble by visualization. Sparse convolution is not suitable for INR, because the positions queried during training do not spatially meet the form of 2D pixel arrays.
>
> ***W14: It is recommended that the ablation study in Tab. 7 be performed more standardized on COCO 128x128 ...***
>
> CrowdPose has higher requirements for heatmap generation. The $\sigma$ of Gaussian distribution should not be set too large, otherwise it will be difficult to distinguish the same kind of joints from different instances when they are close. Therefore, CrowdPose is more worthy of study to determine the scale parameters $\sigma$ and $b$. Thanks for the reviewer's suggestion, we will perform ablation with input size of 128x128.
>
> ***W15: Lack of heatmap visualization and comparisons with other methods.***
>
> Compared with the existing methods with fixed output sizes, NerPE can generate the predicted heatmaps at arbitrary resolutions. Without retraining, visualization of NerPE's outputs at different resolutions is given in **Fig. A2 of the attached PDF**.

---

> ### Comment · Reviewer_xx7e · 2024-08-13
> **Conditional Borderline**
>
> I carefully read the reviews of other reviewers and the author's careful rebuttal. I sincerely thank the author for addressing most of my concerns carefully, especially the experiments on GFLOPs, inference latency of FPS, different input and cell resolutions, SOTA backbones, etc., which help to have a more comprehensive understanding of the proposed method. The idea of ​​INR is very straightforward, especially achieving good performance in low-resolution experiments. So if the author can promise in addition to the rebuttal,
>
> 1. to add comparison and combination experiments with the "offset" method later (see W9 and other reviews),
>
> 2. to add some more visualization of non-standard Gaussian heatmaps, such as the difference between the rendered and baseline heatmaps for medium and hard cases [c] (Ws 10 & 15),
>
> 3. Open source is encouraged to facilitate reproduction for the community.
>
> then I will consider to raise my rating to **borderline** accept (**conditional**).
>
> In addition, there are some follow-up questions:
>
> **W13:** Can you explain why sparse convolution is not considered, which has been explored in 3D reconstruction and generation?

---

> > ### Author Response · Authors · 2024-08-14
> >
> > We sincerely thank the reviewer for the kind reply and willingness to improve the score. We will supplement the experiments related to offset prediction and more visualization of non-standard Gaussian heatmaps in the next/final revision of our paper, as suggested by the reviewer.
> >
> > > Can you explain why sparse convolution is not considered, which has been explored in 3D reconstruction and generation?
> >
> > If we understand correctly, in 3D reconstruction and generation, continuous 3D data (i.e. point clouds) needs to be voxelized into a grid structure to meet the requirements of sparse convolution. This process is similar to heatmap discretization in human pose estimation, which is exactly what we want to avoid. Back to our proposed NerPE, during training, the 2D coordinates as one of the inputs are randomly sampled in the image plane, so they exist in the form of a set in space and cannot be processed by convolution operations. This is why implicit neural representations used to model continuous signals are mostly implemented by MLP even when the task is image-related.

---

### Author Rebuttal · Authors · 2024-08-07

We would like to thank all reviewers for their time and their constructive feedback. We appreciate their assessment of our work NerPE as a "well-motivated" approach for "an important research topic" (xx7e), “a more principled solution to producing arbitrary-resolution heatmaps” (k3b9), a "new idea introduces implicit neural representations" (KNNC), and a "well-justified" approach to "address the weakness of prior works" (rvLc). All reviewers seemed to agree that our work is "good writing" (rvLc), "easy-to-understand" (xx7e, KNNC), and "clearly structured" (k3b9).

Inspired by the reviewers’ helpful comments, we will incorporate the following changes into the next/final revision of our paper:

-  We test the performance of NerPE at a higher resolution (384x288) and with another backbone (ViTPose) on the COCO test-dev set (see Tab. A1 of the attached PDF).

- We report more about the division of cells in NerPE (see Tab. A2 of the attached PDF).

-  We compare the efficiency and performance of NerPE and existing methods (see Tab. A3 of the attached PDF).

-  We visualize the ablation of local resemble in NerPE (see Fig. A1 of the attached PDF).

-  We show the heatmaps output by NerPE at different resolutions (see Fig. A2 of the attached PDF).

---

### Comment · Area_Chair_27DM · 2024-08-10
**Reviewer-author discussions**

Thanks to Reviewer k3b9 for the discussion on the rebuttal.



Dear Reviewer xx7e, KNNC and rvLc,

Would you please look at the rebuttal, discuss with the authors and finalize your score after discussion?

Thanks,
You AC

---

### Decision · Program_Chairs · 2024-09-25

**Decision:**

Accept (poster)

**Comment:**

This paper proposes to use implicit representations to address heatmap quantization error for keypoint localization. It received 2 BAs and 2 WAs after rebuttal, which all the reviewers agreed to accept.


The initial concerns focused on clarification, visualization, and especially insufficient experiments (resolution, backbone, and efficiency). Moreover, some important related works (like heatmap & offset, non-standard Gaussian heatmap, and regression) are missing.

After going through the paper, the review, and the response, the AC values the key idea, finds most concerns are addressed, and recommends the acceptance of the paper.

The AC also recognizes the weaknesses (insufficient experiments and missing related works) and thus urges the authors to further revise the paper.